# Update on the Epidemiological Features and Clinical Implications of Human Papillomavirus Infection (HPV) and Human Immunodeficiency Virus (HIV) Coinfection

**DOI:** 10.3390/microorganisms10051047

**Published:** 2022-05-18

**Authors:** Alexandre Pérez-González, Edward Cachay, Antonio Ocampo, Eva Poveda

**Affiliations:** 1Group of Virology and Pathogenesis, Galicia Sur Health Research Institute (IIS Galicia Sur), 36312 Vigo, Spain; eva.poveda.lopez@sergas.es; 2Infectious Disease Unit, Internal Medicine Department, Hospital Álvaro Cunqueiro, 36312 Vigo, Spain; antonio.ocampo.hermida@sergas.es; 3Division of Infectious Diseases and Global Public Health, Department of Medicine, University of California at San Diego, San Diego, CA 92093, USA; ecachay@health.ucsd.edu

**Keywords:** human papillomavirus (HPV), human immunodeficiency virus (HIV), coinfection, cervix cancer, anal cancer, screening

## Abstract

Human papillomavirus (HPV) infection is the most common sexually transmitted infection (STI) worldwide. Although most HPV infections will spontaneously resolve, a considerable proportion of them will persist, increasing the risk of anogenital dysplasia, especially within certain populations, such as patients infected with human immunodeficiency virus (HIV). Furthermore, high-risk oncogenic HPV types (HR-HPV) are the main cause of cervix and other anogenital cancers, such as cancer of the vagina, vulva, penis, or anus. HIV and HPV coinfection is common among people living with HIV (PLWH) but disproportionally affects men who have sex with men (MSM) for whom the rate of persistent HPV infection and reinfection is noteworthy. The molecular interactions between HIV and HPV, as well as the interplay between both viruses and the immune system, are increasingly being understood. The immune dysfunction induced by HIV infection impairs the rate of HPV clearance and increases its oncogenic risk. Despite the availability of effective antiretroviral therapy (ART), the incidence of several HPV-related cancers is higher in PLWH, and the burden of persistent HPV-related disease has become a significant concern in an aging HIV population. Several public health strategies have been developed to reduce the transmission of HIV and HPV and mitigate the consequences of this type of coinfection. Universal HPV vaccination is the most effective preventive tool to reduce the incidence of HPV disease. In addition, screening programs for HPV-related cervical and vulvovaginal diseases in women are well-recognized strategies to prevent cervical cancer. Similarly, anal dysplasia screening programs are being implemented worldwide for the prevention of anal cancer among PLWH. Herein, the main epidemiological features and clinical implications of HIV and HPV coinfection are reviewed, focusing mainly on the relationship between HIV immune status and HPV-related diseases and the current strategies used to reduce the burden of HPV-related disease.

## 1. Introduction

Human papillomavirus (HPV) is a double-stranded DNA virus transmitted through intimate and sexual contact. HPV is distributed worldwide, and most people in the sexually active population have been exposed to it. HPV is acquired during initial sexual relations, regardless of gender. Most HPV infections spontaneously resolve; however, some medical conditions (i.e., diseases that weaken the immune system) are associated with the persistence of the infection [1]. These medical conditions can be congenitally acquired, such as in the case of epidermodysplasia verruciformis [2], or non-congenitally acquired. The latter category includes people living with human immune deficiency virus (HIV), those with cancer undergoing chemotherapy, chronically immunosuppressed individuals, or those who have undergone organ transplantation [3,4].

HPVs comprise five genera each with the potential to infect humans, although the tropism differs for each genus [5]. In addition, HPV genotypes are also classified based on their oncogenic potential [6], and a fraction of HPV types are related to anogenital dysplasia [3]. HIV and HPV coinfection is common, and the clinical implications of this status are noteworthy. The burden of HPV disease among PLWH is significant, both in men and women. For example, although only a fraction of HPV-infected patients will progress to develop invasive HPV-related cancer, the prevalence of some neoplasms, such as anal squamous cell carcinoma (SCC), is 60- to 80-fold higher among PLWH than those without HIV [1,7].

Worldwide, the most common HPV-related cancer type is cervix SCC, followed by oropharyngeal vaginal, vulvar, penile, and anal SCC [8,9,10]. The trend of HPV-related malignancies varies based on the studied location. Notably, a decrease in cervix SCC is expected because of the scale-up of HPV vaccination among women [11]. Nonetheless, data concerning anal SCC is equivocal, and discrepancies might be detected around the world [12,13]. However, the incidence of anal SCC has increased in PLWH over the last few years, although future projections suggest these figures are stabilizing [13,14,15]. Conversely, an increase in the incidence of HPV-driven oropharyngeal SCC is predicted in the coming years [16]. Vulvar, vaginal, and penile SCC are less common than cervical or anal SCC, and disparity data are available for these diseases [15].

Vaccination has become the main preventive tool for HPV disease, both in men and women, regardless of their HIV status, helping to promote a decline in dysplasia [11,17]. Widespread vaccination has changed the prevalence of high-risk HPV (HR-HPV), both in PLWH and those without HIV, and an accompanying reduction in the prevalence of HPV-16 and HPV-18 has also been observed [18]. Gender-neutral vaccination policies have been successfully developed in several countries (e.g., Australia). However, inclusive vaccination policies for both girls and boys at risk are scarce in some others (e.g., Spain).

This review represents a comprehensive update of the main epidemiological characteristics and clinical consequences for the outcome of HIV–HPV coinfection, both in men and women. We also review the risk factors for HPV-related cancer, the relevance of dysplasia screening programs, and HPV vaccination.

## 2. Methods and Search Strategy

We searched the PubMed database for publications related to clinical trials and observational studies using a combination of the following terms and abbreviations: “HIV”, “HPV”, “HPV vaccine”, “cervix cancer”, “anal cancer”, “vulvar cancer”, “vaginal cancer”, “penile cancer”, “oropharyngeal cancer”, and “sexually transmitted infection”. The literature search started in October 2021 and ended in March 2022, and we included scientific data published between January 1996 and March 2022.

## 3. Human Papillomavirus

HPVs are double-stranded DNA viruses containing 8 genes and over 160 HPV types have been fully characterized [19]. Five genera of HPV are involved in human disease, each with a specific tropism and virulence level [5] (Figure 1). The alpha-papillomavirus (αHPV) genus comprises most of the clinically significant HPV strains, although species from beta-papillomavirus (βHPV) and gamma-papillomavirus (γHPV) genera have also been isolated in PLWH [20]. Mu-papillomavirus (µHPV) and nu-papillomavirus (νHPV) are related to cutaneous diseases (i.e., skin warts) [21]. The HPV genome contains two encapsulating structural proteins, L1 and L2 (Figure 2). 

The L1 protein is the major element of the capsid and is essential for viral binding and entry into human cells [23]. Overexpression of L1 alone is enough to form virus-like particles (VLPs). L1-VLPs can bind to cell surfaces, but L2, the minor capsid protein, is required for cell internalization [24]. The infective HPV virions attach to basal skin stem cells through subtle disruption of the epithelium. Upon entry, virions start a low-level replication cycle and remain in the epithelium basal cells. When these basal cells begin to differentiate through the epithelium stratum, HPV activates a productive phase of the viral life cycle. Finally, once assembled, virions are liberated into the most superficial epithelial stratum [25].

### Tropism and Genotypes

The HPV genera exhibit different tropisms for human structures. Within the αHPV genus, several species prefer the cutaneous epithelium of the hands and feet (i.e., plantar warts or common warts). In contrast, others preferentially infect the anogenital squamocolumnar mucosa or keratinized skin (i.e., cervix, vagina, anal canal, penis, scrotum, or perineum) [26]. In contrast, µHPV and γHPV manifest mainly cutaneous tropism [21], while βHPV and γHPV genera are traditionally associated with cutaneous tropism [27], although species from both have been isolated from the anal canal of PLWH [28,29]. Moreover, HPV species are clinically classified based on their oncogenic risk and relationship with dysplasia (the pre-cancerous state), in four groups: high risk (HR-HPV), probably high risk, low risk (LR-HPV), and indeterminate risk [30] (Table 1).

## 4. Natural History of HPV Infection

Upon entry into the basal epithelial cells, HPV promotes several alterations in cell-cycle regulation through the activity of E6 and E7 proteins. On the one hand, the E6 protein binds p53, a major negative regulator of cell growth. The loss of p53 activity leads to uncontrolled cellular cycling, allowing chromosomal mutations to accumulate without DNA repair [31]. On the other, E7 protein targets p130, another negative cell-cycle regulator, thereby increasing cell growth and promoting uncontrolled proliferation [32]. The loss of cell-cycle control and cumulative DNA mutations leads to the emergence of dysplasia, characterized by keratinization, especially in the latter stages of the disease [33]. During this process, other mutations involving genes implicated in cell-cycle regulation may occur (e.g., in *p53*, *EGFR*, *KRAS*, or *BRAF* [34]), which increase the grade of dysplasia and risk of cancer development. Interestingly, some biomarkers (e.g., DNA methylation or HPV p16 gene integration into host cells) are being investigated as predictors of a higher risk of progression towards invasive anal or cervix cancer [35,36].

The interval between HPV infection and the eruption of squamous intraepithelial lesions (SILs) usually requires several years. At the histological level, the progression of HPV infection correlates with different changes or stages. The initial stages of the disease involve mild or moderate changes in the epithelium with these lesions being termed low-grade squamous intraepithelial lesions (LSILs; Figure 3), of which a considerable proportion spontaneously regress. However, a small number of LSILs may progress further and become high-grade squamous intraepithelial lesions (HSILs; Figure 4 and Figure 5), which are characterized by severe epithelial changes (e.g., intense keratinization and nuclear changes). Although HSILs might progress further and transform into squamous cell carcinoma (SCC) [37], most of them naturally resolve or remain stable. The risk of developing an SCC and the spontaneous regression rate for these lesions is influenced by several factors including HIV infection, the presence of other immunosuppressive diseases, and HPV genotypes.

Both LSILs and HSILs receive specific names based on the affected tissue: CIN, VaIN, VIN, AIN, and PeIN for cervix, vagina, vulva, anus, and penis tissue intraepithelial neoplasia, respectively. They are also traditionally graded into categories based on the severity of the dysplasia (grade 1 for mild, grade 2 for moderate, and grade 3 for severe dysplasia). Although the American Society for Colposcopy and Cervical Pathology recommends only using the terms LSIL and HSIL rather than the traditional three-tier classification (-IN1, -IN2, and -IN3) [38], the system is still used and is considered helpful for clinical management.

Our knowledge of HPV infection and its relationship with dysplasia has dramatically increased over the last few decades, especially regarding cervix dysplasia (although several gaps in our knowledge remain). Indeed, the cervical disease has become a model for HPV-related dysplasia in the context of other anogenital areas, such as the vagina, vulva, or anus. Vaginal and vulvar dysplasia are approximately 100-fold less common than their cervical counterparts [39] although their incidence is higher in women living with HIV [8]. In contrast, anal and penis dysplasia follow a similar disease pattern to cervix disease. Penile dysplasia is unusual and is closely related to HIV infection and anal dysplasia [40]. Conversely, oropharyngeal dysplasia is less well-known, and slightly different oncogenic pathways might be involved in this disease [41,42].

## 5. Relationship between HIV and HPV

The main route for transmission of both HIV and HPV is intimate sexual contact. The molecular interactions between these two viruses are not fully understood, but several mechanisms have been recently proposed. Firstly, in in vitro models, the HIV tat protein transactivates the HPV long control region, increasing the expression of the oncogenes *E6* and *E7* [43]. Moreover, HIV tat protein promotes cell-cycle progression and reduces the expression of cell-cycle inhibitors [44]. Secondly, HIV infection induces an immunosuppressive status by decreasing the levels of CD4+ lymphocytes and impairing the dendritic cell activation and activity of CD8+ lymphocytes. Interestingly, CD8+ lymphocyte activity may play a key role in eliminating HPV-infected epithelial cells [45]. In addition, the infiltration of CD8+ lymphocytes in HPV-related squamous cancer could improve the prognosis of the disease [46,47]. Therefore, HIV activity against both CD4+ and CD8+ lymphocytes may decrease the clearance of HPV-infected epithelial cells and facilitate cell-cycle dysregulation.

### 5.1. Clinical Implications of HIV–HPV Coinfection

Coinfection of HIV and HPV increases the risk of several diseases such as cervical and anal dysplasia. According to the World Health Organization (WHO), 570,000 women were diagnosed with cervical cancer in 2018 worldwide, 5.8% of whom were living with HIV [48]. Indeed, a recent meta-analysis reported an increased pooled risk of cervical cancer in women living with HIV (with a risk ratio (RR) of 6.07) [49]. In addition, the incidence of anal dysplasia is also more common in PLWH, especially among MSM. A pooled analysis of more than 3000 patients reported a higher risk for HSIL in MSM living with HIV (with an adjusted prevalence ratio (PR) of 1.54) [50]. Furthermore, the incidence of several other HPV-related cancers, including oropharynx, vagina, vulva, and penis cancers was increased in PLWH [51]. A low CD4+ lymphocyte count could also be related to a poorer outcome in some HPV-associated neoplasms [52]. The incidence of penile SCC is extremely low, although it is higher in PLWH than in those without HIV [53].

### 5.2. HPV-Related Cancers in People Living with HIV

HPV and HIV coinfection is frequent in men and women, and clearance of HPV infection is less efficient in PLWH compared with the general population [54]. Nonetheless, progression from persistent HPV infection to pre-malignant changes and invasive cancer is slow [55]. Hence, most PLWH coinfected with HPV will not develop cancer, although the overall incidence of HPV-derived invasive cancer in PLWH is higher than in other groups at risk [51] (Figure 6). Furthermore, PLWH on antiretroviral therapy (ART) are now living a longer life, leading some researchers to estimate that, by 2030, the sum of all HPV-related cancers in PLWH will surpass the absolute number of new prostate cancers, with the latter currently being the most common cancer diagnosed in PLWH [11].

Cervix SCC is the most common HPV-related cancer worldwide, with an adjusted incidence ratio in 2020 of slightly over 7 cases per 100,000 inhabitants in high-income countries [56]. Cervical SSC accounted for approximately 300,000 deaths per year but with variable geographic distributions [57]. The incidence of cervix SCC was higher in low-income regions and remained one of the most frequent causes of death in several countries in Africa [58,59]. The incidence of cervical SCC is roughly fourfold higher in women living with HIV compared with their HIV-uninfected counterparts [60]. Vaginal and vulvar dysplasia are approximately 100-fold less common than cervical dysplasia. Notably, the incidence of vaginal SCC has remained stable over the last few decades, accounting for around 8000 cases per year in the United States [61,62]. Vulvar SCC is also uncommon, and most cases are diagnosed in women living with HIV [63,64].

Worldwide, in comparison with other prevalent cancers and the overall population at risk, anal SCC is rare and accounts for approximately 0.4% of all cancers. Globally, 50,865 new cases of anal cancer were diagnosed in 2020, and 19,293 patients died of anal SCC; the largest number of cases occurred in women not infected with HIV [65]. In Spain, the incidence of anal cancer in PLWH increased from 20.7 to 97.8 per 100,000 person years between 2004 and 2017 [14]. Severe immunodepression and a prior AIDS diagnosis were identified as risk factors for anal SCC (adjusted RRs = 2.5 and 10.8, respectively). Overall, anal SCC is more frequent in women than men in the general population (adjusted incidence rate = 0.66 and 0.35 per 100,000 inhabitants, respectively) [65].

Penile cancer is also unusual in the general population, with around 30,000 cases being diagnosed worldwide in 2020 [65]. However, a recent Danish study reported an increase in the incidence of penile SCC from 0.85 to 1.13 cases per 100,000 person years from 1997 to 2017 [66]. Another epidemiological study performed in the Netherlands reported an incidence of 0.47 cases per 100,000 men [67]. Nevertheless, the incidence of penile SCC in PLWH is approximately threefold higher than that in HIV-uninfected men. Several observational studies have reported an incidence rate for penile SCC between 3.4 and 3.8 cases per 100,000 person years, especially in the African American population [68].

Worldwide, 98,000 oropharyngeal SCCs were diagnosed in 2020 [59]. The incidence of oropharynx squamous cell cancer (OPSCC) also increased both in men and women by 0.6% per year in the last decade, reaching 11.2 cases per 100,000 inhabitants in the United States [13]. Although tobacco and alcohol consumption—traditional risk factors for OPSCC—is declining, the incidence of this cancer type is increasing, owing to a higher rate of HPV-related OPSCC. HPV-driven OPSCC tends to affect younger people with no significant smoking or alcohol consumption, with HPV-16 being the genotype most commonly linked to the disease [69]. Finally, the incidence of HPV-related OPSCC is roughly fivefold higher in PLWH than in those without HIV (50.6 cases per 100,000 person years) [52,70].

## 6. Relationship between Antiretroviral Therapy and HPV

ART has become the most useful tool for providing an almost normal life span to PLWH by helping them to maintain an adequate immune status along with the virological suppression of HIV. Here, we examine the impact of ART on the immunovirological status and HPV-related diseases.

### 6.1. Relationship between ART and HPV Prevalence

Several studies have explored the relationship between ART and the prevalence of HPV. However, the evidence available to date is equivocal because of differences in the study designs and different methodologies used in these studies.

### 6.2. Cervical HPV Infection

Numerous researchers have reported a lower prevalence of cervix HPV infection in PLWH receiving ART, compared with untreated patients. A meta-analysis including 6500 women infected by HIV reported a lower HR-HPV prevalence in the pooled analysis than in those not receiving ART (with a crude OR of 0·82) [71]. In addition, an observational study conducted in Kenya also reported that a longer duration of ART was associated with a lower HPV infection prevalence of all HPV genotypes and HR-HPV specifically (with ORs, of 0.90 and 0.87, respectively) [72]. This could reflect better immune control and viral clearance of HPV infection in the cervix among women receiving ART, changes in risky sexual behaviors, or a combination of both.

### 6.3. Anal HPV Infection

Several observational studies have also reported a relationship between ART and a lower risk for anal HR-HPV, both in women and men [73,74]. According to a meta-analysis comprising over 400,000 patients, virologically suppressed PLWH (i.e., those with HIV-RNA under 50 copies/mL) had a 35% lower risk of having HR-HPV infection than ART-naïve PLWH. These results were irrespective of the historical level of immunosuppression according to the nadir CD4 cell count or current CD4 value. Notably, prolonged HIV viral suppression with the use of ART was associated with a 10% reduction in HPV prevalence per year [75]. In addition, a prospective study conducted in Spain observed a reduction in the prevalence of HR-HPV after the initiation of ART (76.6% vs. 59.5%, respectively) but not regarding LR-HPV infection (73.1% vs. 68.8%, respectively) [76].

### 6.4. HPV Infection of the Male Genitals

HPV infection of the male genitals (i.e., glans, coronal sulcus, scrotum, or urethra) was less common than anal infection, ranging between 15% to 55% in PLWH [63,77,78], although some limitations might have influenced these data. For instance, there was no clear standard for sample collection across the studies and the anatomic location examined for HPV detection might have varied. Whether ART contributes to reducing the prevalence of genitalia infection in men remains unknown.

### 6.5. Oral HPV Infection

OPHPV infection in PLWH is less common than cervical and anal infections, but a relatively limited number of studies have been carried out in this field. There is no clear standard for sample collection, although rinsing or the use of a cytologic brush has become the most common method to gather oral samples for HPV detection. Whether ART might reduce the prevalence of HPV infection remains unclear. No significant reduction in oral HPV infection was found 12 weeks after ART initiation in a prospective cohort (18% vs. 24%, respectively), but the sample size was low, and no solid conclusions could be drawn [79]. The major limitations of these publications were their lack of behavioral data and long-term follow-ups. Indeed, it was unclear if changes in the prevalence were reflective of temporary risky behaviors or a biological effect. Moreover, the prevalence of HPV shedding during HPV infection and disease did not correlate with the prevalence of dysplasia among PLWH, as noted below.

### 6.6. Relationship between Antiretroviral Therapy and HPV-Related Dysplasia

The relationship between ART and HPV-related dysplasia is equivocal. A pooled analysis of 14 observational studies did not find evidence of an association between ART and a lower prevalence of HSIL-CIN2+ [71]. However, in the same meta-analysis, a pooled analysis of three studies suggested that the prevalence of CIN2+ was lower in women receiving ART for more than 2 years compared with ART-naïve women. Interestingly, two observational studies focusing on the risk of progression from LSIL to HSIL reported a 34% lower risk for progression to HSIL if ART was initiated before the LSIL diagnosis and a 36% lower risk for progression among women who had been on ART for more than 2 years [80]. In contrast, a nested case–control study on a Swiss cohort reported that ART had a protective effect against CIN2+ (OR ART-experienced vs. ART-naïve = 0.64) but not against invasive carcinoma [81].

Regarding anal dysplasia, whether ART reduces the risk of HSIL or anal cancer remains controversial. During the first 15 years of highly potent ART, the incidence of invasive anal cancer has continued to increase regardless of gender [82]. Furthermore, a recent meta-analysis reported a 44% reduced risk of anal cancer in PLWH with a sustained HIV viral load suppression, albeit with considerable heterogeneity between studies, which possibly limited the results [75]. Moreover, several prospective studies have found a lower prevalence of high-grade anal dysplasia in patients receiving ART than in ART-naïve patients [83,84].

### 6.7. Relationship between the CD4 Lymphocyte Count and HPV Infection

Several studies have reported an increased risk for HPV infection in PLWH receiving more advanced immune suppression such as those affected by AIDS-defining diseases or profound CD4 depletion (i.e., CD4 counts < 200 cells/µL). However, the heterogeneity and cross-sectional design of most of the studies might limit these conclusions. A cross-sectional study completed in China recently reported an increased risk for anal HPV infection in PLWH, especially if the nadir CD4 was lower than 200 cells/μL (OR = 1.80). Moreover, a CD4 lymphocyte count under 350 cells/μL was also related to an increased risk for anal infection (OR = 2.06) [85]. Similarly, a cross-sectional study in Spain reported a lower CD4 lymphocyte count among MSM with an HPV infection [86]. Several observational studies have also reported an increased risk for cervical infection in PLWH, with a baseline CD4 lymphocyte count of < 100 cells/μL [87,88].

Contradictory results have also been reported for oral infections in relation to the potential increased risk for HPV infection in patients with a low CD4 lymphocyte count. [89,90,91]. Similarly, Mooij et al. reported a lack of correlation between the nadir CD4 lymphocyte count and a higher risk for genitalia infection (i.e., penis and scrotum) in a large cohort of MSM [92].

### 6.8. Relationship between the CD4 Lymphocyte Count and HPV-Related Dysplasia

The relationship between a low CD4 baseline or nadir lymphocyte count and cervical dysplasia is still controversial. Although a low CD4 lymphocyte count is related to a higher prevalence of HPV, a relationship between CD4 levels and CIN has not been established. In fact, a prospective cohort comprising 40 women infected with HIV with a follow-up of 2 years did not find any concordance between the CD4 lymphocyte count and CIN1+ [93]. Similarly, previous research has found no correlation between the CD4 lymphocyte count and cervical dysplasia [94]. Conversely, it has been reported that lower CD4 counts increase the risk of vulvar dysplasia [95].

Regarding anal dysplasia, several prospective cohorts have reported a relationship between a low nadir CD4 lymphocyte count and a higher risk for HSIL [96,97]. Indeed, a retrospective study cohort comprising 2800 HIV-infected male or female patients reported a lower risk of progression of LSIL to HSIL among those who presented virological suppression (defined as HIV RNA levels <400 copies/mm^3^) or CD4 lymphocyte counts >350 cells/µL [98]. Moreover, a low CD4/CD8 ratio was related to a higher incidence of HSIL or anal SCC [99] and may be a biomarker for high-risk patients [100]. 

Little is known about the processes involved in oral and penile dysplasia in PLWH or its relationship with HIV, the CD4 lymphocyte count, or ART adherence. Previous studies found no clear correlation between dysplasia and a low nadir CD4 lymphocyte count (i.e., <200 cell/µL), although most of them were affected by limited sample sizes [40,101]. Focusing on oral dysplasia, it is difficult to analyze whether a low CD4 lymphocyte count is related to a higher risk of dysplasia or progression from LSIL to HSIL because there is no systematic screening for oropharyngeal dysplasia. Furthermore, the classification and categorization of oral dysplasia are not well-defined compared with the system used for anogenital processes [42]. Nonetheless, current evidence suggests that immunosuppression and more advanced stages of HIV infection increase the risk of oropharyngeal SCC. In this regard, a nested case–control study reported a higher risk for OPSCC in PLWH with a CD4 lymphocyte count <200 cell/μL (adjusted OR = 2.1) [102]. A retrospective, cohort study comprising more than 40,000 PLWH also reported a higher risk of HPV-related OPSCC in patients with a CD4 lymphocyte count <200 cell/μL [70].

## 7. Dysplasia Screening Programs in People Living with HIV

Over the last few decades, cervical dysplasia screening programs have been developed to help reduce the incidence of invasive cervical cancer, both in women living with HIV and those not infected with the virus. These programs led to a decline in the mortality rates attributed to cervical cancer, especially in high-income countries. Current guidelines recommend performing HPV DNA detection, Papanicolaou (Pap), or visual inspection after acetic acid (VIA) tests, although the former has become the preferred method according to the WHO [48]. Women living with HIV should be routinely screened for cervical dysplasia, especially those with low CD4 lymphocyte count or high HIV viral load [49]. Despite effective ART and adequate immunological control, women living with HIV have a higher risk of progression toward invasive carcinoma compared with those without HIV [103]. Therefore, cervix dysplasia screening programs must be included in the follow-ups of PLWH.

The incidence of anal cancer in PLWH has also increased in recent years, both in men and women. Thus, most health systems, especially in high-income countries, continue to scale up the implementation of anal cancer screening programs, with a similar structure and objectives to those of cervical screening programs [104,105,106]. Elements of anal screening programs include a digital rectal exam, anal cytology, and anal swabs to detect HPV-DNA. Like the colposcopy procedure performed in cervical dysplasia screening programs, the anal mucosa can also be examined directly with an anoscope, and acetic acid impregnations may reveal areas of dysplasia. Treatment modalities include topical imiquimod, infrared coagulation, electrocautery, and surgical excision, according to the type of lesion [107]. The ANCHOR clinical trial recently compared intervention versus observation during anal dysplasia screening. Treatment of anal HSIL decreased the risk of developing anal SCC by 57%, thus reinforcing the role of anal dysplasia screening among PLWH [108].

The incidence of HPV-related penile and oropharyngeal cancer has also increased, but both these dysplasias are less frequent than cervical and anal cancer. To date, no penile or oral dysplasia screening programs have been developed.

## 8. Microbiomes and HPV and HIV

Our knowledge about human microbiomes has improved over the last few decades, although the relationship between microbiomes, the immune system, and several oncogenic viruses such as HPV is not yet fully understood. The emerging evidence highlighting the interaction between different microbiomes and HPV infection is summarized below.

### 8.1. The Cervical Microbiome, HPV, and HIV

The relationship between HPV and the cervical microbiome has widely been studied in recent years. Although notable advances in our understanding have been made, important gaps in our knowledge still remain. Di Paola et al. studied successive cervicovaginal samples obtained in an HPV screening program and created two main study groups: women who cleared HPV and those diagnosed with a persistent HPV infection. At the end of the follow-up, the microbiome detected in the persistent group was rich in anaerobic bacteria of several genera including *Pseudomonas*, *Brevibacterium*, and *Peptostreptococcus*, but *Lactobacillus* was less common [109]. 

Other studies have also reported changes in *Lactobacillus* spp. populations in HPV-infected women [110,111]. In addition, Mengying Wu et al. investigated the cervical microbiome of women diagnosed with LSIL and HSIL versus a control group and reported a higher proportion of *Delftia* genus in the HSIL group, while the *Peptostreptococcaceae* family and *Pseudomonadales* order were more common in the control group [112]. Moreover, although the molecular mechanisms are unknown, there is also increasing evidence to link other bacterial species (*Brevibacterium* spp. and *Brachybacterium* spp.) to cervical dysplasia and cancer [113,114].

### 8.2. The Anal Microbiome and HIV

Previous studies have reported differences in the gut and anal microbiome in MSM living with or without HIV. However, the interactions between the anal microbiome and dysplasia are not yet fully understood. The HIV-uninfected anal microbiome is enriched in *Prevotella* species, while the *Fusobacterium* phylum and *Sneathia* genus are more frequent in HIV-infected microbiomes [115]. In addition, other studies have found the presence of *Ruminococcaceae*, a family of bacteria in the *Clostridia* class, and *Alloprevotella* spp. as predictors of HSIL [116]. *Catenibacterium* (an anaerobic genus) and *Peptostreptococcus* have also been recognized as predictors of anal dysplasia [117].

### 8.3. The Penile Microbiome, Circumcision, and HPV–HIV Coinfection

Previous clinical trials indicate that penis circumcision reduces the risk of both HIV infection (60% protective effect) [118] and HPV infection [119], although the mechanistic cause of this effect is not fully understood. Regarding the penile microbiota and HPV infection, Onywera et al. reported several differences in HPV-infected males, who were enriched in *Prevotella* and *Peptoniphilus* spp. In contrast, *Corynebacterium*, *Micrococcus*, *Sanguibacter*, and *Brevibacterium* spp. were more frequently detected in larger quantities in HPV-infected men than in those not infected with HPV [120]. Thus, some authors have suggested a relationship between *Corynebacterium* spp. and Langerhans cells that facilitates clearance of HPV infection [121,122].

### 8.4. The Oral microbiome, HIV, and HPV

Recent studies have reported several differences in the oral microbiome in HPV-infected and HPV-uninfected patients. The oral microbiome of HPV-infected patients is enriched in *Actinomycetaceae*, *Prevotellaceae*, and *Veillonellaceae* compared with HPV-uninfected patients [123]. In addition, the oral microbiome in patients diagnosed with HPV-related oropharyngeal SCC is richer in anaerobic bacteria than in healthy controls [124]. The detection of STI pathogens such as *Chlamydia trachomatis* or *Neisseria gonorrhoeae* is also associated with lymph node metastasis [125]. However, several factors might have limited the conclusions of these studies, including, in most cases, their cross-sectional design, limited numbers of studied populations, and importantly, the fact that the length of the HPV infection was usually unknown. Hence, more evidence should be gathered in this field to yield potential preventive or therapeutic interventions.

## 9. Sexually Transmitted Infections and HPV and HIV

The incidence of several STIs has increased in recent years [126]. *Treponema pallidum*, *Chlamydia trachomatis* (CT), *Neisseria gonorrhoeae* (NG), and HPV are the most common sexually transmitted pathogens in the United States, and their incidence rate increased by 30% to 60% from 2015 to 2019 [127]. Coinfection of HIV, HPV, and other STIs such as NG or *Mycoplasma genitalium* (MG) is common [128], and the risk of contracting an STI is increased according to the number of sexual partners [129], condomless receptive anal intercourse [130] and drug consumption [131]. 

Epidemiological studies suggest an increased risk of anal dysplasia among PLWH who became infected by CT during their follow-up [132], although the underlying mechanisms of this increase are unknown. A recent study suggested a higher number of centrosomes and multinucleation rate in cells infected by both CT and HPV than in those infected only by HPV [133]. Nonetheless, numerous concerns might limit the results of these investigations. For instance, the length of time between HPV and other pathogenic infections is usually unknown. Moreover, the higher incidence of STI and HPV infection might be owed to high-risk sexual practices in this population (i.e., condomless sexual intercourse, multiple sexual partners, etc.) rather than a mechanistic cause.

## 10. HPV Vaccination in People Living with HIV

Three HPV vaccines have been commercialized worldwide and remain mainstream types of treatment for primary HPV disease prevention, although many others are currently under development. Cervarix^®^ was the first commercialized vaccine and contained VLPs corresponding to HPV-16 and HPV-18. Gardasil-4^®^ was the second developed vaccine and contained VLPs of HPV-6, HPV-11, HPV-16, and HPV-18. Finally, Gardasil-9^®^ expanded coverage to HPV-31, HPV-33, HPV-45, HPV-52, and HPV-58. The bivalent vaccine was commercialized in 2008 and was included in the vaccination schedule of many countries including Scotland for girls aged 12–13 years. 

A corresponding retrospective population study performed 20 years later reported an 89% reduction in preinvasive cervical disease [11]. In addition, long-term follow-up of HPV vaccine clinical trials also reported a nearly 100% incidence reduction in CIN2+ attributable to HPV-16 or HPV-18 [134]. The bivalent vaccine also decreased the risk of HPV infections in women with a prior diagnosis of HPV-16 or HPV-18 infection (vaccine efficacy 68.6% and 81.2%, respectively) [135]. Moreover, long-term follow-up studies suggest that a stronger immune response is induced by the bivalent vaccine than by the quadrivalent vaccine [136]. 

Although first-generation vaccines were mainly assessed in young women (aged 16–26 years), the nonavalent vaccine was also successfully evaluated in older women (27–45 years old) [137]. The immune response in PLWH was similar to that in patients not infected with HIV, both in men [138,139] and women [17,139,140]. The cellular immunogenicity induced by the bivalent or quadrivalent vaccine in PLWH was remarkably similar to that of people without HIV [141], although a more intense humoral immune response was detected in patients who received a bivalent vaccine compared with those receiving a quadrivalent vaccine [142]. In addition, the immune response induced by the quadrivalent vaccine remained stable over time [143].

### 10.1. The Impact of HPV Vaccination on Dysplasia

Previous clinical trials and observational studies showed a reduction in HPV-related dysplasia in vaccinated women [11,144]. Moreover, the HPV vaccine may be helpful as adjuvant therapy for cervical dysplasia to help reduce the risk of recurrence. A recent meta-analysis evaluated 7 studies with a total of 3375 patients and reported a lower rate of CIN1+ recurrence in vaccinated women (6.3% vs. 10.5%, respectively). In addition, the recurrence of CIN1+ attributed to HPV-16 or HPV-18 was also lower in vaccinated women (1.1% vs. 3.1%, respectively) [145]. 

Another meta-analysis comprising more than 2900 previously unvaccinated women diagnosed with cervical dysplasia also reported a lower rate of CIN1+ recurrence in women who received standard dysplasia treatment plus HPV vaccination, both for CIN1+ (RR = 0.35) and CIN2+ (RR = 0.41) [146]. However, the results were not so successful for women living with HIV. In this sense, a clinical trial in South Africa included HIV-infected women previously diagnosed with cervical HSIL who were randomized to receive the HPV quadrivalent vaccine or a placebo. At week 52, the recurrence of HSIL was similar in both arms of the study (32% vs. 31%, respectively) [147].

Regarding anal dysplasia in MSM living with HIV, several clinical trials failed to demonstrate a reduction in the recurrence of HPV-related dysplasia in this population [148,149]. In addition, to date, the relationship between HPV vaccination and the risk for oral and penile dysplasia remains unknown.

### 10.2. Recommendations for HPV Vaccination

Currently, the HPV vaccine is universally recommended for all women, regardless of other risk factors such as immunosuppression. Universal vaccination of women led to a decline in the incidence of cervical disease years later [150,151], as well as lower HPV (HPV-6, HPV-11, HPV-16, and HPV-18) infection rates in MSW, owing to herd protection from female-only vaccination schedules [152]. However, most MSM do not benefit from this herd protection, and therefore, specific vaccination schedules have focused on this population [153]. Indeed, universal vaccination of men, regardless of their sexual behaviors, has been implemented in some countries (including the USA, France, Germany, New Zealand, and Australia) and has led to a substantial reduction in the incidence of genital warts [154]. In Spain, gender-neutral vaccination policies are recommended by the Spanish Association of Pediatrics, although the final decision still lies with each Autonomous Community [155].

## 11. Conclusions

HPV-related diseases are of major concern for both men and women. Coinfection of HIV and HPV is common and is associated with an increased risk of HPV-related cancers. HPV is the main cause of several cancers, including cervical and anal dysplasias. The burden of disease is expected to increase in the forthcoming years because of an increase in the incidence of various HPV-related cancers. The main preventive tool is the HPV vaccine, which should be administered to all children during adolescence. The main role of HPV vaccination is primary prevention, both in men and women. By contrast, secondary prevention has no apparent benefit in PLWH. In addition, adherence to ART, sexual counseling (i.e., advice regarding condom usage), and screening programs are also helpful to reduce the burden of HPV disease.

## Figures and Tables

**Figure 1 microorganisms-10-01047-f001:**
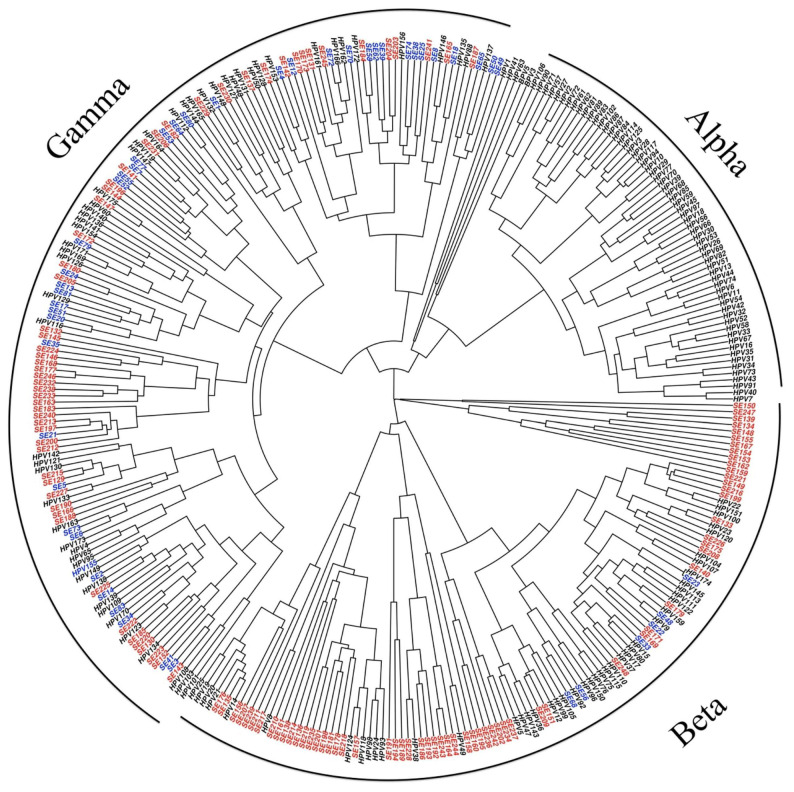
A Bayesian phylogenetic tree based on the L1 gene. From Bzhalava et al. [5].

**Figure 2 microorganisms-10-01047-f002:**
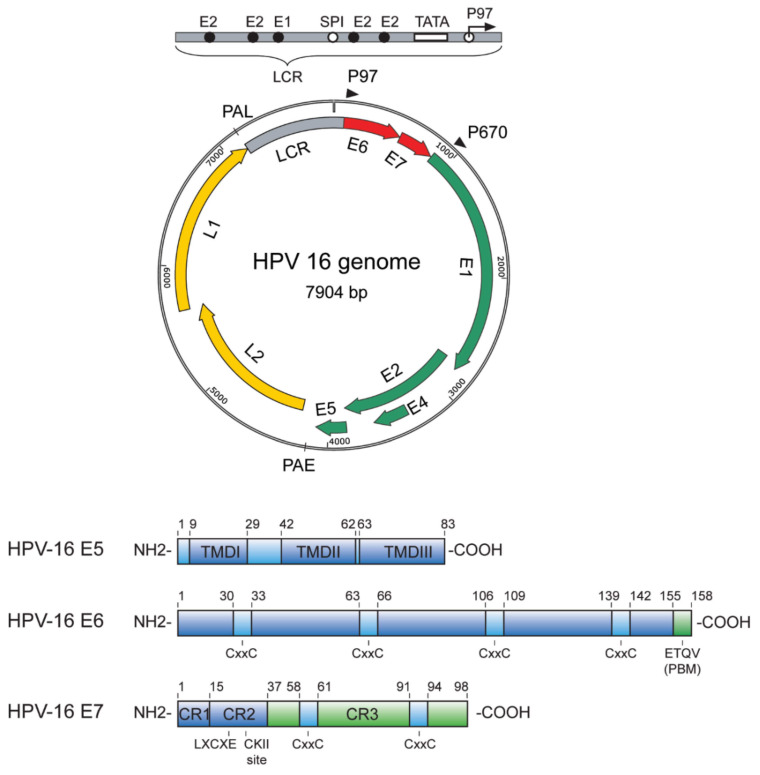
Schematic representation of the HPV genome [22].

**Figure 3 microorganisms-10-01047-f003:**
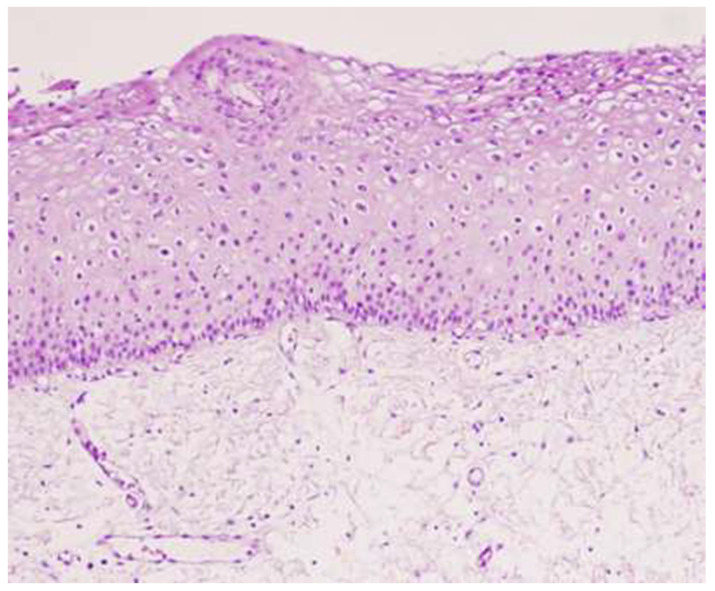
Low-grade squamous intraepithelial lesion (LSIL), 40×. Courtesy of Dr. Mauricio Iribarren, General Surgery Department, Álvaro Cunqueiro Hospital, Vigo.

**Figure 4 microorganisms-10-01047-f004:**
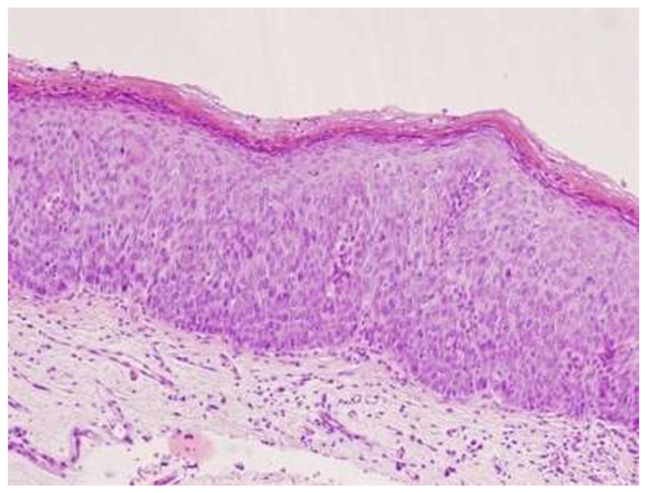
High-grade anal squamous intraepithelial neoplasia (AIN2+), 40×. Courtesy of Dr. Mauricio Iribarren, General Surgery Department, Álvaro Cunqueiro Hospital, Vigo.

**Figure 5 microorganisms-10-01047-f005:**
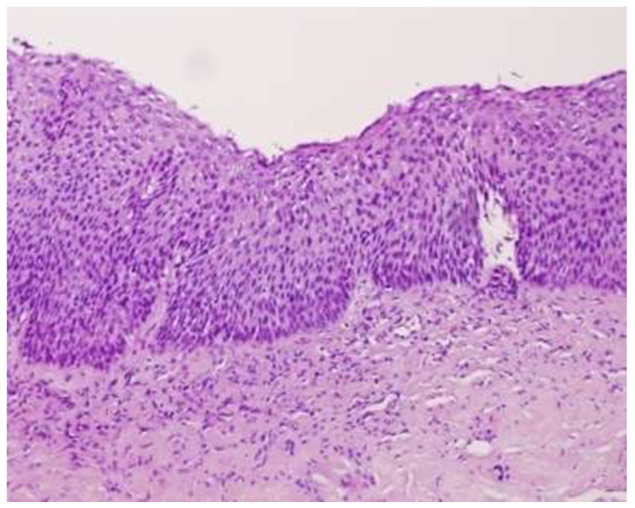
High-grade anal squamous intraepithelial neoplasia (AIN3+) 40×. Courtesy of Dr. Mauricio Iribarren, General Surgery Department, Álvaro Cunqueiro Hospital, Vigo.

**Figure 6 microorganisms-10-01047-f006:**
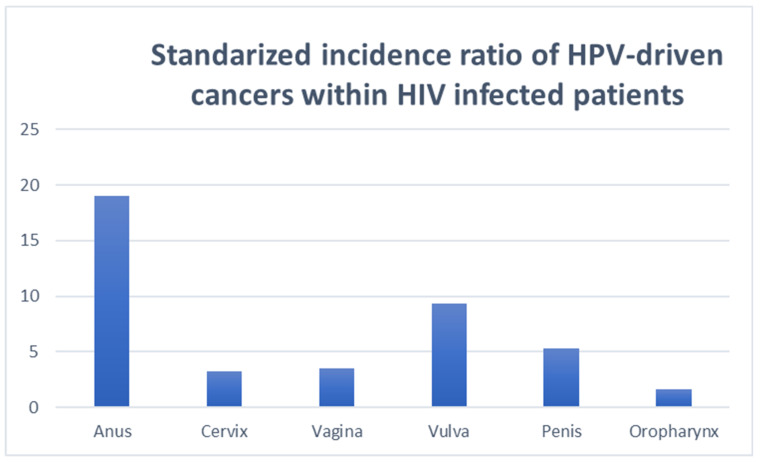
The standardized incidence ratios of HPV-related cancers in people living with HIV.

**Table 1 microorganisms-10-01047-t001:** HPV classification according to the oncogenic risk (IARC).

Classification	HPV Types
High risk (group 1)	16, 18, 31, 33, 35, 39, 45, 51, 52, 56, 58, 59
Probably high risk (group 2A)	68
Possibly carcinogenic to humans (group 2B)	26, 53, 66, 67, 70, 73, 82 30, 34, 69, 85, 97
Unclassifiable as to carcinogenicity in humans (group 3)	6, 11

## Data Availability

Not applicable.

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
