# Peer review of "Update on the Epidemiological Features and Clinical Implications of Human Papillomavirus Infection (HPV) and Human Immunodeficiency Virus (HIV) Coinfection"

_microorganisms, 2022, doi:10.3390/microorganisms10051047_

Round 1

Reviewer 1 Report

The paper describes extensively the epidemiology and clinical aspects of HIV and HPV coinfection. The data presented are interestingly adding a usefull review of the current knowledge.

However, the manuscript suffers from many grammatical and syntax errors, that make difficult its reading.

Also, Figure 1 is not clear at all.

Table 1 and reference 5 need to be updated. They are rather obsolete.

Reviewer 2 Report

The work is an up-to-date review of the literature on HPV and HIV co-infections. The work is very important from a clinical point of view. Noteworthy is a very detailed and extensive review based on the current literature. The authors discuss the immunological aspects of viral infection, the effects on oncogenesis, and also pay attention to other STDs as well as changes in the microbiome and the use of vaccines.
It is worth adding a few sentences of the comment regarding:
- Did the literature review find differences in co-infections over the decades?
- Is the impact of HIV greater / the same / less than other immunosupresive factors (i.e. Wielgos et al. Int. J. Environ. Res. Public Health 2022; Wielgos et al. Viruses 2020)

In addition, it is worth paying attention to minor editorial errors in order to improve readability:
- the font in the abstract is different at the beginning and afterwards
- abbreviations should be developed the first time they appear in the text, regardless of the abstract
Figure 3 is right after Figure 1, while Figure 2 is closer to the end of the text.

Round 2

Reviewer 1 Report

Nothing to suggest